Does haemosporidian infection affect hematological and biochemical profiles of the endangered Black-fronted piping-guan (Aburria jacutinga)?

Motta Rafael Otávio Cançado 1
Romero Marques Marcus Vinícius 2
Ferreira Junior Francisco Carlos 2
Andery Danielle de Assis 2
Horta Rodrigo Santos 1
Peixoto Renata Barbosa 3
Lacorte Gustavo Augusto 1
Moreira Patrícia de Abreu 1
Paes Leme Fabíola de Oliveira 3
Melo Marília Martins 3
Martins Nelson Rodrigo da Silva 2
Braga Érika Martins embraga@icb.ufmg.br 1
1 Departamento de Parasitologia, Instituto de Ciências Biológicas, Universidade Federal de Minas Gerais (UFMG) , Belo Horizonte, MG , Brazil
2 Departamento de Medicina Veterinária Preventiva, Escola de Veterinária, Universidade Federal de Minas Gerais (UFMG) , Belo Horizonte, MG , Brazil
3 Departamento de Clínica e Cirurgia Veterinária, Escola de Veterinária, Universidade Federal de Minas Gerais (UFMG) , Belo Horizonte, MG , Brazil
Kissinger Jessica
Electronic publication date: 2013 Feb 26
Publication date: 2013
Volume: 1
Electronic Location ID: e45
Received 2012 Dec 3; Accepted 2013 Feb 9
Copyright: © 2013 Motta et al.
Copyright year: 2013
Copyright holder: Motta et al.
License: This is an open access article distributed under the terms of the Creative Commons Attribution License, which permits unrestricted use, distribution, and reproduction in any medium, provided the original author and source are credited.
License URL: https://creativecommons.org/licenses/by/3.0/

Keywords: Avian malaria, Plasmodium, Haemoproteus, Captive, Conservation

Funding: Conselho Nacional de Desenvolvimento Científico e Tecnológico (CNPq) Fundação de Amparo a Pesquisa do Estado de Minas Gerais (FAPEMIG) Coordenação de Aperfeiçoamento de Pessoal de Nível Superior (CAPES) Érika M. Braga received funds from Conselho Nacional de Desenvolvimento Científico e Tecnológico (CNPq), Fundação de Amparo a Pesquisa do Estado de Minas Gerais (FAPEMIG) and Coordenação de Aperfeiçoamento de Pessoal de Nível Superior (CAPES). The funders had no role in study design, data collection and analysis, decision to publish, or preparation of the manuscript.

==============================
Infectious diseases can cause deleterious effects on bird species, leading to population decline and extinction. Haemosporidia can be recognized by their negative effects on host fitness, including reproductive success and immune responses. In captivity, outbreaks of haemosporidian infection have been observed in birds in zoos and aviaries. The endemic Brazilian Atlantic rainforest species Aburria jacutinga is one of the most endangered species in the Cracidae family, and wild populations of this species are currently found mainly in conservation areas in only two Brazilian states. In this study, we aimed to evaluate the effects of avian haemosporidia on hematological and biochemical parameters in two captive populations of A. jacutinga. Forty-two animals were assessed, and the haemosporidian prevalence was similar for males and females. The occurrence of haemosporidian infection in captive A. jacutinga observed in this study was similar to results found in other captive and wild birds in Brazil. We found three different lineages of haemosporidia. Two lineages were identified as Plasmodium sp., one of which was previously detected in Europe and Asia, and the other is a new lineage closely related to P. gallinaceum. A new third lineage was identified as Haemoproteus sp. We found no significant differences in hematological and biochemical values between infected and non-infected birds, and the haemosporidian lineage did not seem to have an impact on the clinical and physiological parameters of A. jacutinga. This is the first report on an evaluation of natural haemosporidian infections diagnosed by microscopic and molecular methods in A. jacutinga by hematology, blood biochemistry, and serum protein values. Determining physiological parameters, occurrence and an estimation of the impact of haemosporidia in endangered avian species may contribute to the management of species rehabilitation and conservation.

Introduction

Infectious diseases can greatly impact local species populations by causing temporary or permanent declines in host abundance. The effects of infection interacting with other driving forces, such as habitat loss, climate change, overexploitation, invasive species and environmental pollution, contribute to local and global extinctions (Smith, Acevedo-Whitehouse & Pedersen, 2009). Only recently have the negative effects of parasites on the population dynamics of wildlife been recognized and emerged as a critical issue in the conservation of threatened species (Deem, Karesh & Weisman, 2001; Thompson, Lymbery & Smith, 2010).

For a long time, most avian haemosporidian parasites have been considered harmless in the wild because they appear in clinically healthy birds (Remple, 2004). However, researchers have shown that these parasites can have negative impacts on their hosts, especially during energy-demanding or stressful phases by delaying their arrival to breeding grounds, reducing clutch sizes and nest defense behavior, increasing probability of clutch desertion, reducing hatching and fledging success and weaning nestlings with poorer body conditions (Korpimaki, Hakkarainen & Bennett, 1993; Dulfa, 1996; Hakkarainen et al., 1998; Sanz et al., 2001; Andrezj, 2005; Marzal et al., 2005). In captivity, outbreaks of haemosporidian infection have been observed in domestic and wild birds in zoos and aviaries (Fix et al., 1988; Ferrell et al., 2007; Alley et al., 2008; Olias et al., 2011), including threatened species such as the recent report of severe and acute mortality in the masked Bobwhite quail Colinus virginianus ridgwayi at breeding facilities (Cardona, Ihejirika & McClellan, 2002; Pacheco et al., 2011). All of this evidence has deemed avian haemosporidian infections to be important for studies related to wildlife conservation and management programs in-situ and ex-situ  (Atkinson et al., 1995; Kilpatrick et al., 2006).

Most studies have focused on the need to account for infectious diseases in management plans for threatened species, including those maintained in captivity (Deem, Karesh & Weisman, 2001; Smith, Acevedo-Whitehouse & Pedersen, 2009). These studies recommend improving the understanding of the prevalence and diversity of parasites found in such species as a first step, mainly in populations at risk from several other stressors. In addition to surveillance, health assessment of the populations, including baseline information about physiological parameters such as blood count, serum biochemistry profiles and mineral levels and evidence of exposure to residues of chemical contaminants, was also considered essential (Deem, Karesh & Weisman, 2001).

The Cracidae family of the order Galliformes contains 11 genera, 50 species and approximately 60 subspecies (Del Hoyo, 1994). Black-fronted piping-guan, Aburria jacutinga, which is endemic to the Brazilian Atlantic rainforest, is considered one of the most threatened bird species on the American continent. The species was widely distributed at altitudes between 0 and 1000 m and could be found from the northeast to the south of Brazil and in some areas of Argentina and Paraguay. Aburria jacutinga is facing the risk of extinction due to predatory hunting and habitat loss. The Brazilian Atlantic rainforest has been altered and reduced due to deforestation for agribusiness purposes. No reports of the species have occurred in the last 20 years in any of the Brazilian states. Wild populations of A. jacutinga are at present found mainly in conservation areas in only two Brazilian states (MMA, 2008; IUCN, 2010).

To verify the effects of avian haemosporidian infection on the physiological status of A. jacutinga, we assessed the hematological and biochemical parameters in two captive populations. We also present information about the diversity of haemosporidian parasites found in A. jacutinga and provide helpful remarks for the management of this threatened species.

Material & Methods

Bird sampling

This study was conducted in two conservation facilities in the Southeastern region of Brazil (Minas Gerais state): (1) CRAX – Wildlife Research Society (CRAX), (19°51′05″S, 44°04′03″W) and (2) Poços de Caldas Scientific & Conservationist Breeder (CPC) (21°46′59″S, 46°37′24″W). Birds were housed in pairs and enclosures were made of wire chain net. Standard management procedures included feeding once daily (manufactured ration with no antibiotics or anticoccidials) and water ad libitum.

Forty-two blood samples of sexually mature captive A. jacutinga, 11 males and 10 females from CRAX and 8 males and 13 females from CPC, were collected. Birds were captured with a net and restrained manually. Venous puncture and blood collection were performed within five minutes of capture to minimize hematological and biochemical changes caused by the capture stress. Sterile disposable needles (25 G) attached to 3 mL syringes were used for ulnar vein collection. Approximately 3 mL of blood were collected per bird, three blood smears were prepared and 20 µL were stored at room temperature in microtubes containing 300 µL of cell lysis solution (Promega, MA, EUA) for DNA extraction. The remaining blood was immediately transferred into new sterile tubes, with and without heparin, and refrigerated at 4 °C.

The study was approved by the Ethics Committee in Animal Experimentation (CETEA), Universidade Federal de Minas Gerais, Brazil (Protocol #254/2011) and bird samples were authorized by the Instituto Brasileiro do Meio Ambiente e dos Recursos Naturais Renováveis – IBAMA (Number 16359-3).

Microscopic analysis

Smears were air-dried, fixed in absolute methanol and stained for 20 min in 10% Giemsa stain (Sigma Chemical Co., St. Louis, Missouri, USA), pH 7.4 (Ribeiro et al., 2005). Slides were examined by light microscopy (Olympus, Japan) at low magnification (×400) for 10–15 min. Then, 200 microscopic fields (approximately 150 erythrocytes/field) were studied at high magnification (×1000), and parasite densities were quantified, calculating the relative numbers of infected and the total erythrocytes (Ribeiro et al., 2005).

DNA extraction from blood was performed using the Wizard Genomic DNA Purification kit (Promega, MA, USA) according to the manufacturer’s protocols. The DNA pellet was resuspended in 30 µL of hydration solution and stored at −20 °C until use.

DNA samples were initially screened for Plasmodium/Haemoproteus infection by PCR using primers 343F (GCTCACGCATCGCTTCT) and 496R (GACCGG-TCATTTTCTTTG) for the amplification of the structural 18S rRNA gene (Fallon, Ricklefs & Swanson, 2003). The positive controls for PCR were derived from P. gallinaceum genomic DNA (collected from experimentally infected chicks provided by the Laboratório de Entomologia Médica do Centro de Pesquisa René Rachou – CPqRR – Belo Horizonte). The negative control DNA samples were obtained from infection-free chicks (Veterinary College, Universidade Federal de Minas Gerais).

Subsequent to parasite screening, a 524 bp fragment of the mtDNA cytochome b (cyt-b) gene from the infected individuals was amplified by a nested-PCR using primers HaemNFI (CATATATTAAGAGAAITATGGAG) and HaemNR3 (ATAGAAAGATAAGAAATACCATTC) in a first amplification and HaemF (ATGGTGCTTTCGATATATGCATG) and HaemR2 (GCATTATCTGGATGTGATAATGGT) in a second amplification (Hellgren, Waldenström & Bensch, 2004). The PCR products were purified using a solution of 20% polyethylene-glycol 8000 and 2.5 M NaCl (Sambrook & Russell, 2001). After purification, the PCR products were sequenced in both directions using the BigDye Terminator Kit v3 (Applied Biosystems, Foster City, USA) using an ABI3100® automated sequencer (Applied Biosystems, Foster City, USA).

The sequences were assembled and checked for quality using Phred v.0.20425 (Ewing et al., 1998; Ewing & Green, 1998) and Phrap v.0.990319 (Green, 1994). The assembled chromatograms were carefully checked and edited using Consed 12.0 (Gordon, Abajian & Green, 1998). Sequences were aligned using the ClustalW algorithm implemented in MEGA v.5 (Tamura et al., 2011). The genus of each lineage was inferred by the closest sequence matches in Genbank using NCBI nucleotide BLAST search. In an attempt to assign the sequences to described parasite lineages, we compared the sequences with the records in Genbank and MalAvi (Bensch, Hellgren & Pérez-Tris, 2009), which contains cyt-b data for most of published avian haemosporidian parasite lineages. Observed lineages that were not present in the MalAvi database were considered new lineages.

Bayesian analyses were implemented to infer the phylogenetic relationships among cyt-b lineages and morphospecies, whose cyt-b sequences are available in MalAvi database. MrBayes version 3.1.2 (Huelsenbeck & Ronquist, 2001) was used to run simultaneously. Two Markov chains were run for 3 million generations and were sampled every 100 generations. The first 7500 trees (25%) were discarded as “burn-in” and the remaining trees were used to calculate posterior probabilities. Plasmodium and Haemoproteus lineages were analyzed separately because there is no consensus about the monophyly of these genera (Martinsen, Perkins & Schall, 2008; Outlaw & Ricklefs, 2011). A P. gallinaceum sequence was used as an outgroup for the Haemoproteus tree, and H. columbae was used as outgroup for the Plasmodium tree.

Blood parameters analyses

Blood collected in the heparin-containing tubes were stored refrigerated at 4 °C, homogenized and used for total erythrocyte, leukocyte and thrombocyte counts, determination of hemoglobin concentration (Hb) and hematocrit (PCV). All the tests were performed within a maximum period of 4 h after blood collection. The total counts of erythrocytes, leukocytes and thrombocytes were manually performed using a Neubauer chamber as described previously (Natt & Herrick, 1952). Cellular counts were made in duplicate and averaged. Hemoglobin content was determined by the cyanmethemoglobin method using Drabkin solution and spectrophotometer reading at a wavelength of 540 nm. The hematocrit was estimated according to the technique described by Campbell (1991).

Blood samples stored in additive free tubes were centrifuged in 1,238 × g/5 min for serum separation and stored in microtubes at −20 °C. The biochemical tests were performed on the day immediately following collection. The total serum levels of protein (TSP), glucose, uric acid (UA), creatine kinase (CK), aspartate aminotransferase (AST), alanine aminotransferase (ALT), γ-glutamyltransferase (GGT), amylase, creatinine, urea, calcium, phosphorus, alkaline phosphatase (ALP), cholesterol and triglycerides were measured using the Cobas Mira Classic automatic analyzer (Roche Diagnostic) at the Laboratório de Patologia Clínica da Escola de Veterinária da UFMG, using reagents according to the manufacturer (Synermed International Inc., USA).

Total serum protein concentrations were measured by refractometry (Ningbo Utech International Co., model 301, China), and protein fractions, albumin and globulin, were obtained by electrophoresis using an agarose gel electrophoresis system (CELM SDS-60, Brazil). Briefly, a 1 µL serum sample was applied to the gel, which was exposed to 100 V for 30 min, stained (Ponceau S, VETEC Química Fina LTDA, Brazil), fixed and dried. Bands were scanned and quantified using Celm SE-250 image analysis software (CELM – Cia Equipadora de Laboratórios Modernos, Brazil). Absolute values (g/dL) for the protein fractions were determined on the basis of total protein concentration.

The mean and standard deviation (SD) of each hematological and biochemical parameter was calculated according to gender and infection status. The Kolmogorov–Smirnoff test was used to determine whether the data have a normal distribution, and a t-test was used for comparing the means of infected and non-infected animals. The Chi-square test was used to compare the proportions obtained for the different parameters, comparing presence or absence of infection by avian haemosporidia. Statistical analyses were performed with Prism for Windows v.4.0. Differences were considered significant at P < 0.05.

Results and Discussion

All 42 animals studied (19 males and 23 females) appeared to be in good condition, and no abnormalities were detected during the physical examination. The occurrence of Haemosporidian infection was similar for males and females in each breeder. Six males (54.55%) and three females (30%) were infected in the CRAX breeder whereas four males (50%) and five female birds (38.46%) were infected in the CPC breeder. No significant difference in infection rate was noted between breeders, which enabled the grouping of animals according to gender. The overall prevalence of infection reached 52.63% among males and 34.78% for females without gender differences. Regarding the intensity of infection, all animals presented low parasitemias, with less than 1% infected erythrocytes.

We obtained cyt-b sequences of haemosporidia from 10 A. jacutinga, most likely due to the very low parasitemias, which decreases the PCR performance. We identified two different Plasmodium lineages, one of which (ABJAC02) was isolated from five birds and was quite similar to a cyt-b sequence described for P. gallinaceum. The other Plasmodium lineage (JF411406) was isolated from one bird and was previously described in mosquitoes collected in Japan and Turkey (Kim & Tsuda, 2010; Inci et al., 2012). One new Haemoproteus lineage (ABJAC01) was obtained from four birds. The inferred phylogenetic relationships among the morphospecies, whose sequences are available in MalAvi and the Plasmodium lineages found in A. jacutinga, revealed that these two lineages are not closely related, being placed in distinct sub-clades of the tree (Fig. 1). The Haemoproteus lineage found in this study was placed in the Parahaemoproteus clade and constituted a distinct sub-clade on the Haemoproteus phylogenetic tree (Fig. 2). The Plasmodium lineage ABJAC02 was found in five specimens from only one breeder (CPC), and the other two lineages were described on the other facility (CRAX). The diversity of haemosporidian lineages found in the present study is important to meet the demand for genetic variability knowledge of these haemoparasites in the Neotropical region (Braga et al., 2011).

Figure 1 Phylogenetic relationships of the Plasmodium cyt-b lineages found in Aburria jacutinga from two captive populations.

Figure 2 Phylogenetic relationships of the Haemoproteus cyt-b lineages found in Aburria jacutinga from two captive populations.

This is the first screen for avian haemosporidian infection in an endangered species in South America, the Brazilian Atlantic rainforest hotspot. The occurrence of infection in captive A. jacutinga observed in this study was similar to results found in other captive and wild birds in Brazil (Ribeiro et al., 2005; Fecchio, Marini & Braga, 2007; Belo et al., 2009; Belo et al., 2011; Fecchio et al., 2011). The combination of microscopic and molecular analyses ensured a more accurate diagnosis. Indeed, both methodologies were used for haemosporidian diagnosis in this study.

There were no significant differences in hematological values between infected and non-infected males (Table 1). However, infected females showed higher monocyte counts than their non-infected counterparts. We also found no significant differences in biochemical values between infected and non-infected females (Table 2). However, non-infected males presented higher values of LDH (P > 0.05) when compared to infected individuals. Moreover, no significant differences were observed in the serum protein electrophoresis values between infected and non-infected A. jacutinga males or females (P > 0.05, Table 3).

Table 1 Comparison of hematological parameters (mean ± SD) among malaria infected and non-infected captive Aburria jacutinga.

	Male (n = 19)		Female (n = 23)		
	Negative	Positive	P	Negative	Positive	P	
PCV %	38.67 ± 1.73	39.90 ± 5.26	0.5119	38.21 ± 4.30	35.50 ± 3.55	0.1463	
Hb g/dL	13.31 ± 2.22	12.78 ± 1.71	0.5681	12.54 ± 1.20	11.32 ± 1.67	0.1433	
RBC/mm3 × 106	2.243 ± 0.492	2.329 ± 0.531	0.7223	2.095 ± 0.460	1.984 ± 0.369	0.5681	
Thrombocytes/mm3 × 103	6.771 ± 2.401	5.271 ± 1.038	0.0927	5.967 ± 1.541	5.995 ± 2.683	0.9749	
WBC/mm3 × 10	8.104 ± 3.469	7.679 ± 2.373	0.7617	8.855 ± 3.358	12.412 ± 5.561	0.0742	
Lymphocytes/mm3 × 103	5.884 ± 2.421	5.499 ± 2.187	0.7280	6.530 ± 2.906	9.027 ± 4.449	0.1256	
Heterophils/mm3 × 103	1.607 ± 0.617	1.705 ± 0.641	0.7460	1.758 ± 0.695	2.505 ± 1.636	0.1478	
Eosinophils/mm3 × 103	0.337 ± 0.277	0.318 ± 0.218	0.8765	0.338 ± 0.308	0.517 ± 0.402	0.2557	
Monocytes/mm3 × 103	0.134 ± 0.094	0.183 ± 0.163	0.4665	0.141 ± 0.123	0.333 ± 0.160	0.0114	
Basophils/mm3 × 103	0.018 ± 0.05	0.014 ± 0.042	0.8708	0.086 ± 0.193	0.030 ± 0.057	0.4336	

Table 2 Comparison of blood biochemical parameters (mean ± SD) among malaria infected and non-infected captive Aburria jacutinga.

	Male (n = 19)		Female (n = 23)		
	Negative	Positive	P	Negative	Positive	P	
Glucose g/dL	308.00 ± 59.46	290.10 ± 45.02	0.4577	287.79 ± 31.34	284.12 ± 26.40	0.7838	
CK U/L	2801.2 ± 2319.9	2043.6 ± 891.5	0.3478	1828.3 ± 749.0	2482.4 ± 1389.9	0.1635	
AST U/L	117.50 ± 49.53	118.30 ± 49.41	0.9716	116.14 ± 48.53	128.00 ± 40.75	0.5670	
ALT U/L	105.70 ± 23.35	93.20 ± 21.84	0.2529	94.07 ± 32.62	108.87 ± 30.69	0.3085	
GGT U/L	5.79 ± 4.91	5.20 ± 5.31	0.7994	5.35 ± 3.89	4.74 ± 2.79	0.7010	
Amylase U/L	10549 ± 1948.7	12674 ± 3701.9	0.1256	10978.6 ± 2395.6	11466.2 ± 2168.4	0.6402	
Uric Acid mg/dL	14.70 ± 5.48	16.45 ± 6.42	0.5203	16.16 ± 7.08	15.97 ± 6.79	0.9518	
Creatinine mg/dL	0.360 ± 0.184	0.370 ± 0.189	0.9058	0.350 ± 0.285	0.325 ± 0.225	0.8339	
Urea mg/dL	7.90 ± 5.36	7.80 ± 3.01	0.9596	8.64 ± 3.30	7.87 ± 4.121	0.6361	
Calcium mg/dL	10.80 ± 1.03	10.90 ± 1.31	0.8514	11.51 ± 3.23	11.30 ± 1.18	0.8595	
Phosphorus mg/dL	7.87 ± 3.50	7.75 ± 3.41	0.9390	8.29 ± 2.81	7.75 ± 2.55	0.6619	
ALP U/L	266.57 ± 105.50	228.37 ± 92.75	0.4686	228.90 ± 65.40	257.43 ± 113.75	0.5206	
LDH U/L	207.00 ± 48.20	123.80 ± 68.38	0.0201	207.50 ± 79.33	191.40 ± 134.00	0.6839	
Cholesterol mg/dL	170.03 ± 24.50	200.80 ± 23.83	0.0629	172.60 ± 30.37	155.20 ± 25.41	0.1541	
Triglycerides mg/dL	232.00 ± 162.35	203.00 ± 162.35	0.7047	265.40 ± 210.97	247.29 ± 223.97	0.8673	

Table 3 Comparison of serum protein values (mean ± SD) among malaria infected and non-infected captive Aburria jacutinga.

	Male (n = 19)		Female (n = 23)		
	Negative	Positive	P	Negative	Positive	P	
Total Protein g/dL	4.30 ± 0.57	4.25 ± 0.57	0.8464	4.32 ± 0.67	4.19 ± 0.72	0.6650	
Pre-albumin g/dL	0.309 ± 0.081	0.282 ± 0.086	0.496	0.263 ± 0.094	0.286 ± 0.094	0.584	
Albulmin g/dL	2.728 ± 0.294	2.619 ± 0.380	0.498	2.742 ± 0.489	2.714 ± 0.380	0.895	
α-globulin g/dL	0.483 ± 0.240	0.479 ± 0.196	0.966	0.472 ± 0.230	0.411 ± 0.239	0.555	
α1-globulin g/dL	0.342 ± 0.161	0.300 ± 0.137	0.547	0.266 ± 0.115	0.233 ± 0.116	0.532	
α2-globulin g/dL	0.141 ± 0.081	0.179 ± 0.076	0.310	0.206 ± 0.146	0.177 ± 0.146	0.653	
β-globulin g/dL	0.598 ± 0.191	0.751 ± 0.252	0.160	0.724 ± 0.276	0.628 ± 0.244	0.422	
β1-globulin g/dL	0.286 ± 0.211	0.367 ± 0.253	0.525	0.251 ± 0.115	0.292 ± 0.234	0.609	
β2-globulin g/dL	0.335 ± 0.068	0.384 ± 0.129	0.411	0.485 ± 0.270	0.336 ± 0.040	0.140	
γ-globulin g/dL	0.150 ± 0.061	0.120 ± 0.049	0.255	0.140 ± 0.062	0.145 ± 0.267	0.854	

Monocytes play an important role in phagocytic activity and antigen processing in birds. Although monocytosis is usually associated with infection by microorganisms that cause granulomatous inflammation (Campbell, 2004), it has not been correlated to haemosporidian infections. Despite this, a significant difference in monocyte counts was observed between infected and non-infected females, both values are within the expected for healthy Galliformes (Samour, 2006). However, the role of monocytes in avian haemosporidian infections is not well known and requires more study to be fully understood. Another observation was related to the significant difference in the LDH values found between infected and non-infected males. This result seems to be nonspecific because the averages of both groups were within the normal values for LDH previously described in healthy birds (Campbell, 2004). During routine medical evaluation, LDH may be used as a marker of tissue breakdown, as it is abundant in red blood cells and can indicate hemolysis. Thus a false positive result may be obtained for individuals in which blood collection resulted in erythrocyte damage (Campbell, 2004). Seven distinct fractions of serum proteins: pre-albumin, albumin, α1-globulin, α2-globulin, β1-globulin, β2-globulin and γ-globulin were evaluated. The observed values for pre-albumin and albumin were higher than those described previously for Gallus gallus domesticus (Hasegawa et al., 2002), despite that similar values for both species were observed for α-globulin, β-globulin and γ-globulin. Subclinical haemosporidian infection did not seem to cause changes in the electrophoretic profile of A. jacutinga serum proteins, in contrast to a previous report (Williams, 2005) that described a reduction in albumin and α2-globulin and an increase of γ1 and γ2-globulins in experimentally infected Gallus gallus with P. gallinaceum during peak parasitemia.

Previous studies have demonstrated the negative effects of avian haemosporidia on host health (Atkinson & van Riper III, 1991; Desser & Bennett, 1993), including hemolytic anemia, leukocytosis and lymphocytosis as the primary changes in the blood (Campbell, 2004). Increases in plasma total protein, AST, GGT, and glutamate dehydrogenase, as well as a decrease in creatinine values, have also been demonstrated (Williams, 2005). The magnitude of changes seemed to be directly related to the intensity of the infection as measured by the parasitemia (Booth & Elliott, 2002). Most haemosporidian infections in wild birds are subclinical and of low intensity, with rare reports of epizootics associated with captivity and abnormal host-parasite association (Atkinson, 2008). The absence of deleterious effects in A. jacutinga may be a result of the low level of infection, as detected by the low parasitemia, ranging from 1 to 5 parasites per 200 microscopic immersion fields, with few haemosporidian life stages. Due to these low parasitemias, morphological differentiation among Plasmodium and Haemoproteus would be inaccurate, as trophozoites of the former genera could be misidentified as young gametocyte forms of the latter genera and vice versa. The near absence of developing life stages in healthy animals suggests an immune response mediating the controlled infection (van Riper III, Atkinson & Seed, 1994). Absence of effects due to avian haemosporidian infection has been detected in many other avian species (Booth & Elliott, 2002), in Turdus migratorius and T. grayi (Ricklefs & Sheldon, 2007) and in chickens (Nazifi et al., 2008).

Although no major changes were found in the hematological, biochemical or serum protein profiles, basic information regarding aspects of clinically healthy A. jacutinga were obtained. Such information is of interest as there are little data available about normal blood values of endangered species (Deem, Karesh & Weisman, 2001). However, the evaluation of the presence as well as the effects of avian haemosporidians on the health of endangered species, such as A. jacutinga, is still essential because, even though clinical signs of haemosporidian infection are mild or absent in the chronic phase, some studies have shown that haemosporidian infections that are associated with other etiological agents, especially viruses, can amplify the effects of the parasitism (Miller et al., 2001; Davidar & Morton, 2006; Marzal et al., 2008).

Aburria jacutinga is an endangered cracid species with a scarcity of information concerning aspects of its physiology and health, mainly due to the high difficulty of finding individuals both in their natural habitat and in captivity. The presence of undescribed Plasmodium and Haemoproteus lineages in A. jacutinga, indicates the need to characterize the haemosporidian diversity occurring not only in endangered species such A. jacutinga but also in all captive birds maintained in rehabilitation centers or zoos, as well the wild birds that inhabit the vicinity of these centers, to identify potential sources of infection. Estimating the host range of parasites that infect endangered species is also crucial. Parasites that are able to infect multiple hosts coupled with cross species transmission pose the greatest threat to disease-mediated extinction because parasites can maintain high prevalence in alternative hosts even if the density of host populations is very low (Fenton & Pedersen, 2005; Pedersen et al., 2007).

The Brazilian management plan for conservation of cracids proposes reintroduction of endangered species that were born in captivity into the wild (ICMBio, 2008). Therefore, knowledge about physiological parameters and etiological agents that occur in such birds would help to establish guidelines for conservation acts. Considering that the occurrence of haemosporidian parasites was determined and might serve as reference, cracid collections should be evaluated in quarantine and biosecurity should be implemented to establish a health program (Atkinson, 2008). In conservation sites with concentrations of birds, the possibility of a severe challenge to naïve birds in the presence of vectors exists (Ferrell et al., 2007). Thus, using insect-proof nets, at least in enclosures that receive young birds, would reduce the impacts of the disease in early cracid life stages. Additionally, birds to be reintroduced should be assessed for infectious and parasitic etiologies to avoid the introduction of diseases in previously absent environments (Woodford, 2000).

Any conservational action may have parasitological consequences. For instance, programs for reintroducing captive-bred animals that do not take infection risks into account could compromise such programs (Lebarbenchon et al., 2007). The conservation of A. jacutinga species might depend on the evaluation of the impact of the diseases and other threats such as food availability, predation and competition (Kilpatrick et al., 2006). As the chronically infected individuals do survive and reproduce normally, they can be used for breeding in captivity for reintroduction projects and should be assessed for infectious and parasitic agents.

Conclusions

Although some differences in hematological and biochemical parameters had been detected, there is no clear evidence of negative effects due to haemosporidian infection in captive A. jacutinga. However, determining the physiological parameters, occurrence and estimation of the impact of haemosporidia in endangered avian species may contribute to the management in rehabilitation and conservation of species.

We thank the administrators of the CRAX – Wildlife Research Society and CPC – Poços de Caldas Scientific & Conservationist Breeder for allowing sample collection.

Additional Information and Declarations

Competing Interests

Author Contributions

Animal Ethics

Field Study Permissions

DNA Deposition

Érika M. Braga is an Academic Editor for PeerJ. Otherwise, there are no competing interests.

Rafael Otávio Cançado Motta and Marcus Vinícius Romero Marques conceived and designed the experiments, performed the experiments, analyzed the data, wrote the paper.

Francisco Carlos Ferreira Junior performed the experiments, analyzed the data, wrote the paper.

Danielle de Assis Andery, Rodrigo Santos Horta and Renata Barbosa Peixoto performed the experiments.

Gustavo Augusto Lacorte and Patrícia de Abreu Moreira analyzed the data, wrote the paper.

Fabíola de Oliveira Paes Leme and Marília Martins Melo analyzed the data, contributed reagents/materials/analysis tools.

Nelson Rodrigo da Silva Martins conceived and designed the experiments, contributed reagents/materials/analysis tools.

Érika Martins Braga conceived and designed the experiments, analyzed the data, contributed reagents/materials/analysis tools, wrote the paper.

The following information was supplied relating to ethical approvals (i.e. approving body and any reference numbers):

Ethics Commitee in Animal Experimentation (CETEA), Universidade Federal de Minas Gerais, Brazil (Protocol #254/2011)

The following information was supplied relating to ethical approvals (i.e. approving body and any reference numbers):

Instituto Brasileiro do Meio Ambiente e dos Recursos Naturais Renováveis - IBAMA

Number 16359-3

The following information was supplied regarding the deposition of DNA sequences:

The new sequences were submitted to GenBank database and the accession numbers are:

KC250002 (ABJAC01)

KC250003 (ABJAC02)

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
