# Peer review of "Does haemosporidian infection affect hematological and biochemical profiles of the endangered Black-fronted piping-guan (Aburria jacutinga)?"

_PeerJ, doi:10.7717/peerj.45_

## Round 0.1 · original submission · Minor Revisions

I would like to issue a personal thank you for submitting to PeerJ. I have a few comments of my own that I would like to add.

Please consider changing the title to read "Do haemosporidians effect haematological and biochemical profiles in the endangered Black-fronted piping-guan (Aburria jacuntinga)? or "Does haemosporidian infection effect ..."

The manuscript could also benefit from a review of the English by a native speaker.

Please use a prime symbol to indicate DNA sequence orientation rather than an apostrophe.

Reviewer 1 ·

Basic reporting

The manuscript adhered to the formatting and PeerJ policies, is well written with a fine introduction. Figures and tables are appropriate and the manuscript is a fine “unit of publication”.

Experimental design

The manuscript complies with all requirements of PerrJ.

Validity of the findings

The data fits the PerrJ instrucitions

Additional comments

The work by Motta et al. describes the occurrence of haemosporidians in the endangered Atlantic rainforest bird Aburria jacutinga. The presence of blood parasites were search in blood smears of 42 captivity birds; 18 presented haemoparasites. From these 18 birds, molecular analysis revealed, in 10 animals, three distinct sequences indicating the presence of two Plasmodium spp. and one Haemoproteus sp.
The work is scientific sound, is relevant for: a) the understanding of the diversity of haemoparasites in birds, b) the management of endangered vertebrates, and c) hematological parameters of the Aburria jacutinga.

Page 10, line 5. Morphological analysis of blood films found parasites in 18 birds. However, “cyt b” sequences were obtained in only 10 birds and three distinct parasites were found. By the description of the molecular identification of the parasites all 10 birds had monoinfections (lines 6-11). The ABJACO2 was found in five birds, the JF411406 in one and the ABJACO1 in four. Was there any correlation of the morphology of the parasites with the molecular result? Although it is reasoned (page 12, lines 20-22) that morphological differentiation among the parasites would be inaccurate due to parasitemias below 1%, can at least images of the parasites be included to better describe the parasites? In the other 8 birds were sequences were not obtained was there birds with double, or triple infection? Was the morphology of the parasites found in these 8 birds similar? The group has great experience with molecular and morphology techniques. I believe that the correlation between these two characterizations is a must and should be re-evaluated.

Page 11, line 12. The hematological characterization found differences in two parameters: female monocytes counts and male LDH values. The authors consider the LDH result nonspecific, but the difference in monocytes counts seems to be relevant. The parasitemia is described as low, but a real value was obtained, but not informed. The correlation of the parasitemia of individual birds to the hematological results could show that indeed the former is relevant and the latter is not.


Minor suggestions:

Page 8. line 18. Change “1.238 x g/5min” to “1,238 x g for 5min”.
Page 9, line 1. Check two “,” at the middle of the sentence.
Page 10, line 1. Delete “(P>0.05)” at non-significant differences, same for line 3, and page 11, lines 6, 8 and 11.
Page 10, line 7. Move “(ABJACO2)” between “one of them” and “was isolated from five”.
Line 10. Move “(JF411406)” between “lineage”, at line 8, and “was isolated from one” at line 9.
Line 11. Move “(ABJAC01)” between “lineage”, at line 10, and “was obtained”.
Page 11, line 7. Move citation of “Table 1” to end of the sentence at line 6.
Line 9. Add “(P>0.05)” after “values of LDH”.
Page 12, line 12. Use abbreviation for “aspartate aminotransferase” and “gamma-glutamyltransferase”.
Page 14, line 23. Check the use of the word “malaria” at this sentence.
Titles of tables 1, 2 and 3. Use the specie name as for the legends of figures 1 and 2.

Reviewer 2 ·

Basic reporting

The article by Motta et al. asks whether found avian Plasmodium and Haemoproteus lineages have a negative effect on blood profiles of the Black-fronted piping-guan (Aburria jacutinga). The manuscript is well written and includes sufficient background information for a broader audience.

Experimental design

The experimental design seems technically sound.

Validity of the findings

No significant negative effects of the infection on the parameters tested have been found. Therefore the study is merely a description of physiological blood parameters of this particular bird species and the title of the manuscript is misleading and remains unanswered. This is because only a very low infection rate of red blood cells has been found in positive animals and it cannot be excluded that haemosporidians have a negative effect on the health of the investigated bird species i.e. after acute infection. Furthermore, only for ten of 18 positive birds the infection could be verified by PCR although PCR is generally regarded to be more sensitive than microscopical analysis. It would be interesting to compare the results with other related avian species and by this enlarge the number of studied birds. In summary, the data provided is not sufficient to exclude a negative effect haemosporidians on the black-fronted piping-guan.

---

## Round 0.2 · accepted · Accept

Thank you for addressing the reviewer comments and correcting the English.